**RESEARCH**                                                                                    **Open Access**

# Fueling ab initio folding with marine metagenomics enables structure and function predictions of new protein families

Yan Wang[1,2†], Qiang Shi[1†], Pengshuo Yang[1†], Chengxin Zhang[2†], S. M. Mortuza[2], Zhidong Xue[1*], Kang Ning[1*] and Yang Zhang[2,3*]

## Abstract

**Introduction:** The ocean microbiome represents one of the largest microbiomes and produces nearly half of the primary energy on the planet through photosynthesis or chemosynthesis. Using recent advances in marine genomics, we explore new applications of oceanic metagenomes for protein structure and function prediction.

**Results:** By processing 1.3 TB of high-quality reads from the *Tara* Oceans data, we obtain 97 million non-redundant genes. Of the 5721 Pfam families that lack experimental structures, 2801 have at least one member associated with the oceanic metagenomics dataset. We apply C-QUARK, a deep-learning contact-guided ab initio structure prediction pipeline, to model 27 families, where 20 are predicted to have a reliable fold with estimated template modeling score (TM-score) at least 0.5. Detailed analyses reveal that the abundance of microbial genera in the ocean is highly correlated to the frequency of occurrence in the modeled Pfam families, suggesting the significant role of the *Tara* Oceans genomes in the contact-map prediction and subsequent ab initio folding simulations. Of interesting note, PF15461, which has a majority of members coming from ocean-related bacteria, is identified as an important photosynthetic protein by structure-based function annotations. The pipeline is extended to a set of 417 Pfam families, built on the combination of *Tara* with other metagenomics datasets, which results in 235 families with an estimated TM-score over 0.5.

**Conclusions:** These results demonstrate a new avenue to improve the capacity of protein structure and function modeling through marine metagenomics, especially for difficult proteins with few homologous sequences.

## Introduction

To deduce biological functions of proteins, especially for those that are newly discovered but yet have solved structure, computer-based structure prediction can play important roles [1–3]. Two types of modeling strategies have been widely considered for the structure prediction problem [4]. First, template-based modeling (TBM), which constructs structural models using solved structures in the Protein Data Bank (PDB) as templates,

represents one of the most reliable approaches when close homologous templates are detected. The modeling accuracy, however, sharply reduces when the homology level of templates decreases (typically with a sequence identity to the query < 30%) [5]. Therefore, template-free modeling (TFM) approach (or ab initio modeling) has attracted considerable interests in modeling the "hard" proteins that do not have close homologs in the PDB. Due to the lack of reliable long-range atomic interactions in the force field, however, the success rate of traditional physics-based TFM approaches is low and the best approaches can only predict models with limited accuracy for small proteins roughly below 100 amino acids until a few years ago as shown in the community-wide blind CASP tests [6, 7].

Recent CASP experiments have witnessed significant progress in TFM [8, 9], which are mainly attributed to

* Correspondence: zdxue@hust.edu.cn; ningkang@hust.edu.cn; zhng@umich.edu
†Yan Wang, Qiang Shi, Pengshuo Yang and Chengxin Zhang contributed equally to this work.
[1]College of Life Science and Technology and College of Software, Huazhong University of Science and Technology, Wuhan 430074, Hubei, China
[2]Department of Computational Medicine and Bioinformatics, University of Michigan, Ann Arbor, MI 48109, USA
Full list of author information is available at the end of the article

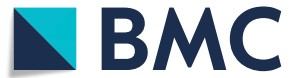

the accuracy improvement of sequence-based contact-map predictions [10–15], as well as efficient coupling of the contact-maps with state-of-the-art structure assembly simulation approaches [16, 17]. Since most contact-map predictions rely on the co-evolution information derived from multiple sequence alignments (MSA) of homologous proteins, the sufficient number of homologous sequences is critical to the success of the approaches. Recently, Ovchinnikov et al. used the Integrated Microbial Genomes (IMG) database [18], which consists of nearly 4M unique sequence entries, to generate contact-map predictions and construct high-confidence models based on Rosetta for 614 Pfam protein families that lack homologous structures in the PDB [19]. Using UniRef20, Michel et al. combined contact-map prediction with the CNS-based folding method to predict the protein structures for 558 Pfam families of unknown structure with an estimated 90% specificity [20].

Despite the remarkable success, most of the approaches generated contact-maps using unified sequence databases from microbial genomes (IMG) or UniProt. Specific genomes from the ocean microbiome, one of the richest sources of organisms on this planet, have recently attracted considerable attention [21, 22]. In particular, structures and functions of the ocean microbiome have very little overlap with the human and animal proteins [23]. However, the specific impact of the ocean-oriented microbial genomes on the contact-map and protein structure and function predictions remain to be examined. Meanwhile, the Gremlin method [24] used by Ovchinnikov et al. for contact prediction is built on co-evolution coupling analysis (CCA), which works reasonably well when the number of homologous sequences is high, but the performance reduces sharply for the sequences lacking sufficient homologous sequences. To partly address this issue, Michel et al. adopted PconsC3, which combines CCA with random forest training [25]. Most recently, the deep-learning-based approach has found significant usefulness for further improving the contact-map prediction accuracy [12, 26], where the accuracy of long-range contact-maps, which are particularly important for 3D structure assembly, increases by nearly twice when coupling co-evolution matrices with deep convolutional neural networks, compared to the CCA-based approaches [27].

In this work, we developed a new pipeline to integrate C-QUARK with the marine microbiome sequences from the *Tara* Oceans database [28] to examine the ability of cutting-edge TFM approaches on genome-wide structure modeling and function annotations, with a focus on the specific impact of the ocean microbiome on the selective Pfam families. Here, C-QUARK is a new ab initio structural assembly method that combines QUARK [29] with contact-map predictions from multiple state-of-the-art contact predictors. In the most recent CASP13 experiment, C-QUARK generated

correct fold (with a TM-score > 0.5) for 33 out of 45 FM and FM/TBM domains, which represents the highest folding rate of FM targets among all automated servers in the experiment [30]. One of the advantages of C-QUARK lies at the ability of QUARK simulations which can fold many sequences with low- to medium-quality models even without the assistance of templates and contact-map predictions [31, 32]. Our study shows that the integration of C-QUARK with deep-learning-based contact-map prediction built on the new *Tara* Oceans databases can significantly increase the yields of computational structural predictions, especially for the non-homologous hard targets, which can benefit the interpretation of functional insights of many protein families that are not accessible from previous approaches and data resources.

## Results and discussions
### Ocean microbiome data processing
To predict genes in the *Tara* Oceans dataset (EBI with project number ERP001736, which contained metagenomic sequencing data for prokaryotic organisms), a volume of 1.3 TB high-quality raw reads were obtained and assembled to 135,132,178 high-quality contigs (N50 length is 982 bps). Based on the assembled contigs, open reading frames (ORFs) were predicted and clustered with 95% sequence similarity, resulting in 97,315,162 non-redundant genes (average length is 426 base pairs).

To examine the microbial community composition, 37,286 microbial operational taxonomic units (OTUs) were obtained, including 816 archaeal and 36,470 bacterial OTUs, respectively. After splitting the taxonomical annotation of the OTUs to genus level, the top four most abundant genera are *Synechococcus* (8.58% ± 7.63%), *Prochlorococcus* (3.13% ± 1.18%), *Candidatus Portiera* (2.76% ± 1.01%), and *Nitrospina* (2.02% ± 1.28%), where all the four genera are dominant genera in marine microbial community [33, 34] (Fig. 1a). Detailed taxonomical distributions on phylum and genus level are listed in Additional file 1: Figure S1 in Supplementary Information (SI). Compared to the IMG database collected on Feb 21, 2017, in which 52.31% of all 17,054 samples are from human and animal gut microbiome (Additional file 1: Table S1), the data in the *Tara* Oceans represent a distinct genetic resource, which can be used for improving protein structure modeling by providing deeper and ocean-microbiome-enhanced multiple sequence alignments with enriched co-evolution information.

### Ab initio protein structure prediction built on *Tara* genome sequences
There are currently 17,929 Pfam families in the Pfam database, where 5721 of them have no member with an experimentally solved structure [35]. Using hmmsearch, 2801 out of the 5721 Pfam families can have at least one member associated with the sequences in the *Tara*

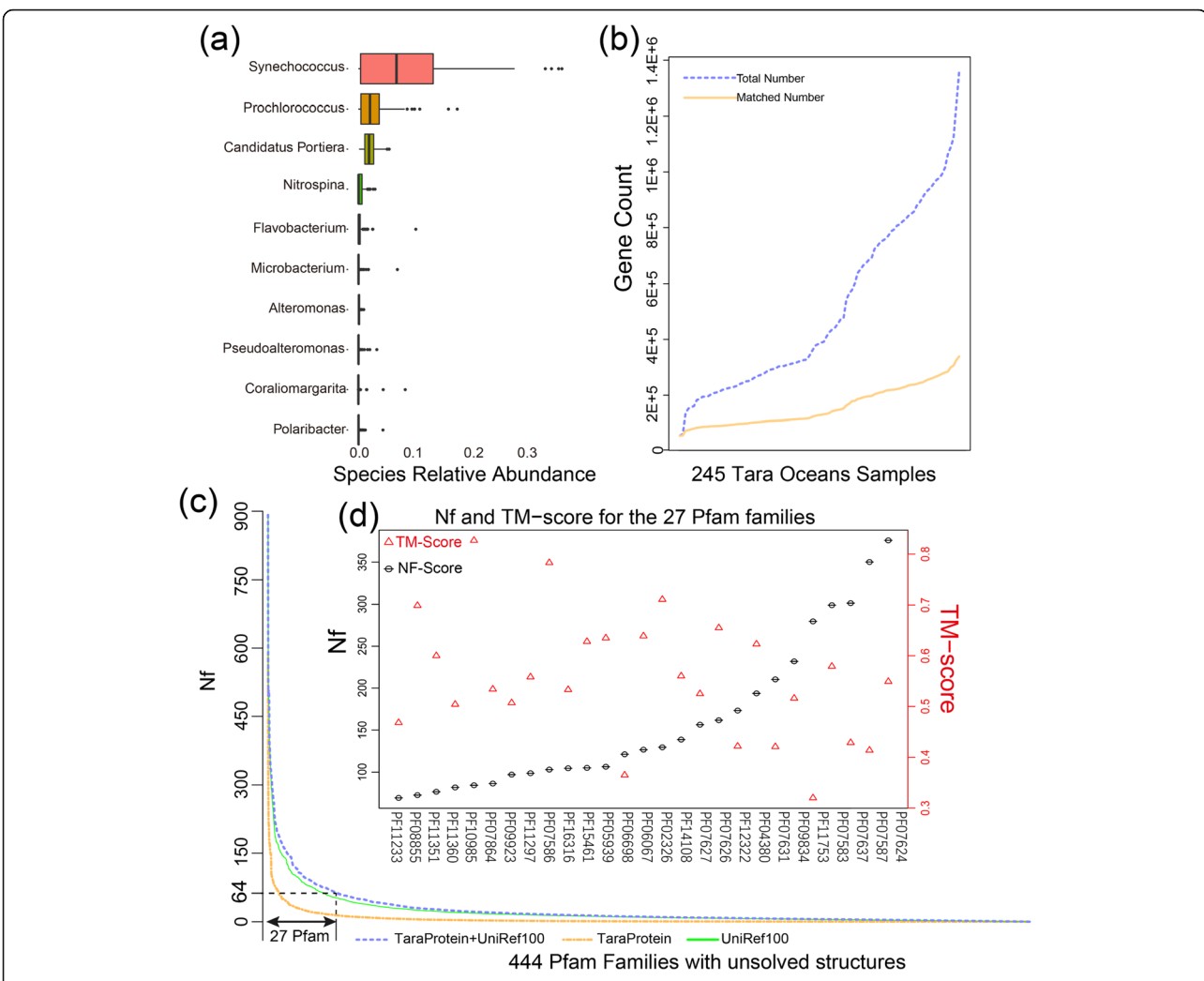

**Fig. 1** Prediction result of unknown protein structure families assisted by marine microbial data. **a** Predicted gene count distribution for the 245 whole-genome sequence runs. Gene distributions predicted from 245 runs of *Tara* data are illustrated and marked as red lines. After assigning these genes to 2801 Pfam families, the assigned gene distribution for the 245 runs are illustrated and marked as green lines. **b** Microbial community profiles of the top 10 genera at the genus level. Vertical axis represents the relative abundance for each genus and horizontal axis the 245 *Tara* ocean samples ranked (from small to large) by their respective gene counts. **c** Nf distribution for the 27 Pfam families. Vertical axis represents the Nf values for the 27 Pfam families. The Pfam with the largest Nf is PF06698 (736), and the smallest NF score is PF11351 (15). Horizontal axis represents the 444 Pfam families ranked by their Nf score. **d** Nf and TM-score distribution for the 27 Pfam families (Nf over 64). Vertical axis represents the NF for the 27 Pfam families. The Pfam with the largest Nf is PF07624 (376), and the smallest Nf is PF11233 (69)

Oceans dataset with an $E$ value < 0.01 (Additional file 2: Table S2 and Fig. 1b), suggesting that the *Tara* genome data can impact the MSAs (and thus the sequence-based contact-map prediction) for nearly half of the structurally unknown Pfam families.

To have a more quantitative assessment of the impact, we reconstructed MSAs for all the 2801 Pfam families based on a combined dataset of the UniRef100, which contains all sequence records from the Uniprot [36], with the *Tara* Oceans dataset. Additional file 2: Table S3 lists the number of effective sequences in each MSA, i.e., $\mathrm{Nf} = \frac{1}{\sqrt{L}}\sum_{i=1}^{n_{\mathrm{seq}}} 1 / [1 + \sum_{j=1, j\neq i}^{n_{\mathrm{seq}}} I(S_{ij} \geq 0.8)]$, where $L$ is the length of the target Pfam protein and $n_{\mathrm{seq}}$ is the total

number of sequences in the MSA, $S_{ij}$ is the sequence identity between $i$th and $j$th sequences in the MSA, and $I(\cdot)$ is the Iverson bracket which equals to 1 if $S_{ij} \geq 0.8$, or 0 otherwise. Nf has been shown to be highly correlated with the accuracy of contact-map predictions and the success rate of the subsequent contact-guided ab initio protein structure prediction [19, 37, 38]. It is shown in Additional file 2: Table S3 that the search through the combined *Tara*/UniRef database resulted in 757 Pfam families having a Nf > 64 (Fig. 1c), a cutoff previously used for successful contact-map prediction [19]; among them, 313 Pfam families have at least one member with structure reported by the authors [19]. The remaining

444 Pfam families were promoted to this category mainly because of the inclusion of the *Tara* Oceans dataset.

In Additional file 1: Figure S2, we show a head-to-head comparison of Nf values of all the 2801 Pfam families calculated on the *Tara* Ocean and IMG datasets that are combined separately with UniRef100. Although the use of IMG generally results in higher Nf values, there are a considerable number of cases (201) in which Nf > 64 occurs only when the *Tara* Oceans dataset is used, indicating the complementarity of the *Tara* Oceans to the IMG dataset for assisting the modeling of different PFam families. Here, we note that the selection of the Nf cutoff (= 64) is empirical. In fact, as per analysis of 45 FM domains in CASP13, although our pipeline can generate correct fold with TM-score > 0.5 for several targets with Nf < 64, the overall quality and success rate for the targets with Nf > 64 are much higher [38]. Approximately, a cutoff of Nf = 64 splits the FM targets into two groups with the average TM-score (0.49 vs. 0.67) that corresponds to the lowest *p* value (= 0.001) in Student's *t* test. We therefore continue to use this cutoff, consistent with the previous study of Ovchinnikov et al. [19].

If we define the Nf fraction due to *Tara* as Nff = ( $\text{Nf}_{\text{Tara + UniRef}} - \text{Nf}_{\text{UniRef}})/\text{Nf}_{\text{Tara + UniRef}}$, we found 27 out of the 444 Pfam families that have a Nff > 0.5, meaning that more than half of the effective sequences are contributed from *Tara* for these proteins (Fig. 1d, or Additional file 1: Figure S3 for a breakdown of all the Pfam families to the 27 entries). The test by the multiple-threading program LOMETS [39] showed that 25 out of the 27 Pfam families are categorized as "hard" targets, suggesting that structure for majority of these families cannot be modeled by the traditional template-based approaches. Figure 2 presents the 3D structures for the representative sequence of the 27 Pfam families modeled by C-QUARK, which were built on the contact-map predictions from the MSAs constructed from the combined *Tara* Oceans metagenomic sequences. Due to the high Nf values (= 158 on average, Additional file 1: Table S4), the confidence of the contact predictions, in particular that of for the top-*L* long-range contacts used by C-QUARK, is high. Accordingly, 20 out of the 27 targets should have a correct fold with a TM-score > 0.5, as estimated by the confidence score of the C-QUARK simulations (see Eqs. (2) and (3) in the "Materials and methods" section). In the remaining 7 targets, 5 have a reasonable estimated TM-score in [0.4–0.5]. Apparently, the *Tara* genome sequences play a particularly important role in modeling these Pfam families which are mostly non-homologous hard targets and with few sequence homologs before the introduction of *Tara* sequence database.

## Functional interpretation of the protein families modeled with *Tara* genomes

The 27 Pfam families modeled with the *Tara* genome dataset belong to 235 genera according to the taxonomy analyses (Fig. 3a). Additional file 2: Table S5 gives a list of the frequency for the 235 genera appearing in the 27 Pfam families. Among these genera, 65% genera belong to *Proteobacteria*, which is a dominate phylum in environment (rather than in host-associated environment) [23]. In addition, several families are from viruses, which is consistent with the observation that marine viruses play important roles as a driver of marine geochemical cycles [40]; these results highlight again the impact of the introduction of the ocean genome sequences on the structural model results. As shown in Additional file 2: Table S5, the genus with the highest occurrences is *Synechococcus* (detected in 25 families), which is a common member in ocean microbiome and annotated as carrying out photosynthesis function [41]. Additionally, 88% of genera appeared only in 1 or 2 families (Fig. 3b), indicating the heterogeneity in the composition of different families. In Fig. 3c, we present a phylogenetic tree for the 27 most popular genera that appear in more than 6 Pfam families, where 21 genera belong to bacteria, 3 to Eukarya, and other 3 to virus (Fig. 3e).

In Fig. 3f, we list the abundance distribution of the 21 bacteria prokaryotes in the *Tara* Oceans dataset. Interestingly, the genera with a high abundance also showed a high frequency of occurrences in 27 Pfam families. For example, the genera *Synechococcous* and *Prochlococcus* have the highest abundance of 8.58% and 3.13% in the *Tara* Oceans dataset, which have also the highest frequency of occurrences in 25 and 14 Pfam families, respectively. This partially explains why the *Tara* Oceans samples are most useful for the MSA and structure predictions on these 27 selected Pfam families. In Additional file 1: Figure S4, we present a comparison of the taxonomical distributions between two protein family sets, one from the 614 modellable Pfam families selected by Ovchinnikov et al. built on the IMG data [19] and another from the 27 Pfam families enhanced with the *Tara* Oceans data. It was shown that the bacteria genomes, which are common in the gut microbiome, account for the overwhelming majority in the former, but the latter is dominated by bacteria genomes from the ocean microbiome. This data demonstrates again the impact of different sequence genomes, as well as the complementarity of *Tara* Oceans with the IMG datasets, on the ab initio protein structure and function predictions.

Additional file 1: Table S6 lists the functional annotation results for the 27 Pfam families using the MetaGO algorithm [42] based on the predicted structures. In Fig. 3d, we tabulated the functions of the Pfam families

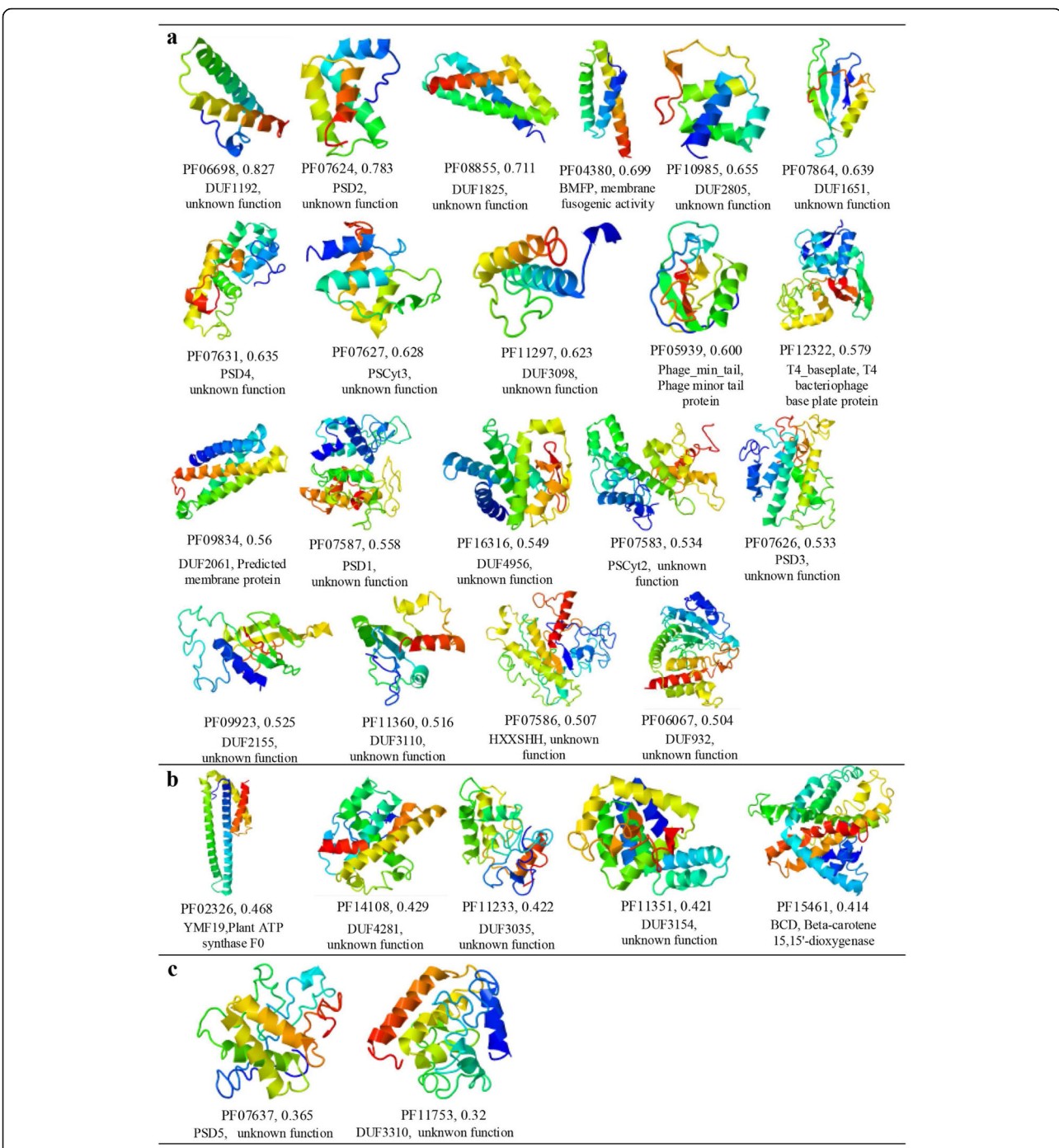

**Fig. 2** C-QUARK models for 27 Pfam families grouped by the estimated TM-score range. Below each of the 3D model, Pfam family number, family name, estimated TM-score, and functional description are listed sequentially. **a** The Pfam families with TM-scores > 0.5. **b** Families with TM-score in [0.4–0.5]. **c** Families with TM-score < 0.4. Out of all 27 predicted structures, 21 are from protein families with unknown functions

in the 27 most popular genera (ranked by the frequency of occurrence in the 27 families), in which the functional aspects of Biological Process (BP), Cellular Component (CC), and Molecular Function (MF) are split into 6, 7, and 5 subcategories respectively. Based on the CC annotation results, the most common function is Integral Component of

Membrane, to which 9 out of 27 families were categorized. Based on MF, Hydrolase Activity (8 families) is the most popular function, which is an important enzyme in ocean microbiome [23]. Among all GO terms observed, two categorizations with the largest number of Pfam families assigned were Metabolic Process (6 families) and Cellular

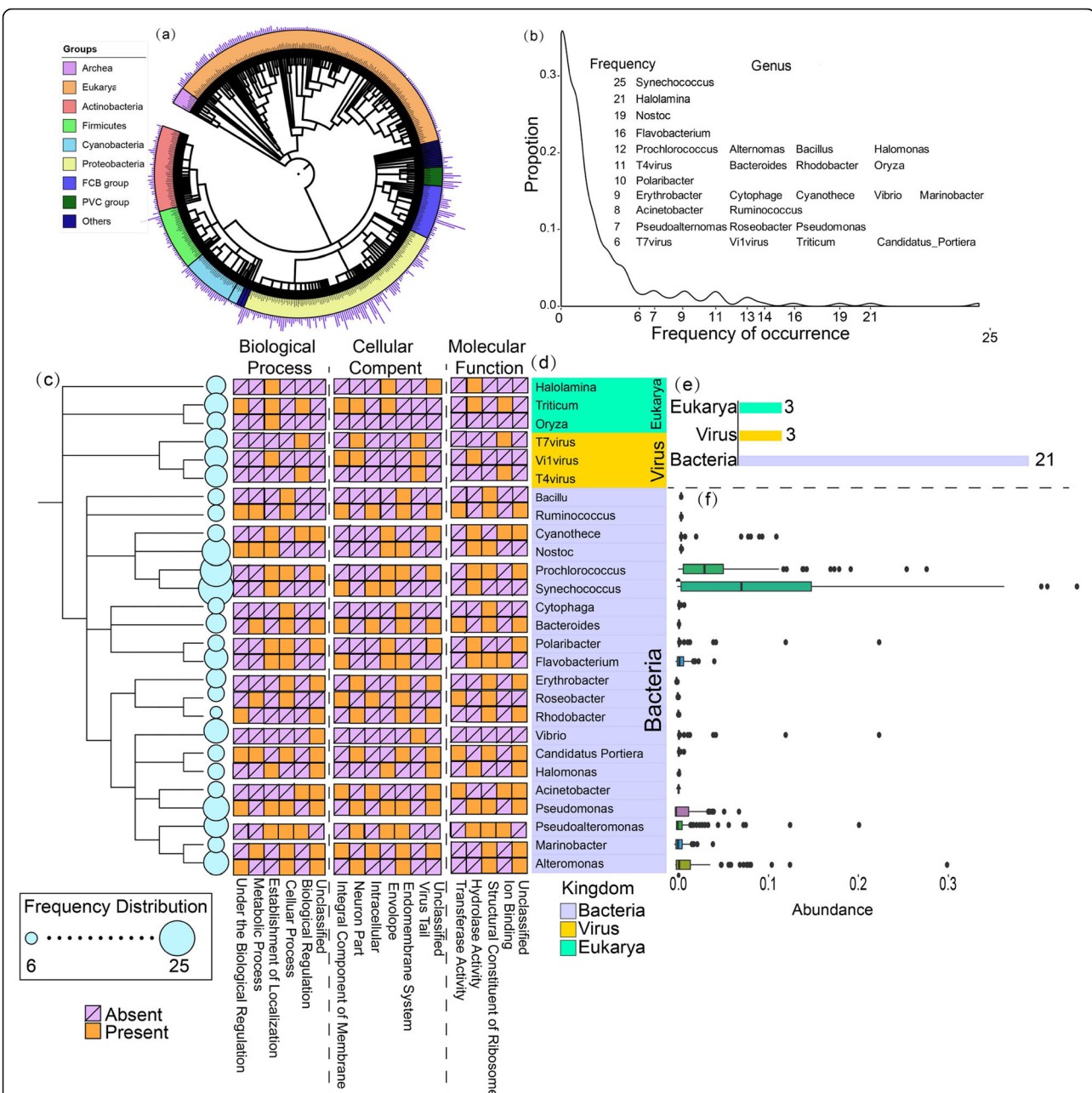

**Fig. 3** Species distribution and structure-based function annotations for the 27 unknown Pfam families. **a** Taxonomical distribution of all the 235 genera. Different colors represent different classifications, and the bar corresponding to the outer circle indicates frequency of the corresponding genera in more than 300 families. **b** Frequency distribution of all the 235 genera in the 27 Pfam families. The vertical axis represents the percentage of species with a specific frequency in 27 Pfam families. **c** A phylogenetic tree of 27 detected genera occurred in over 6 families. The circle size is proportional to the frequency of the species observed from these samples. **d** Function distribution for the 27 genera. The GO functions in Biological Process, Cellular Component, and Molecular Function were classified into 6, 7, and 5 sub-categories, where the sub-category is marked in red if the function was detected in the corresponding genera. **e** Taxonomical distribution of 27 genera. In 27 genera, 3 genera belong to Eukarya, 3 genera belong to virus, and 21 genera belong to bacteria. **f** Relative abundance distribution of 21 bacterial genera in *Tara* Oceans dataset. The relative abundance of 27 genera is calculated and horizontally aligned to corresponded genera

Process (4 families); both are parents of photosynthesis (GO:0015979) which is a GO term that has a high number of Pfam families assigned (167 families in Pfam database). It is also worth noting that photosynthesis (GO:0015979) is a

critical and enriched function for marine microbiomes [23], compared to other biotypes.

To further explore the role of the selected Pfam families in the marine microorganism, in Fig. 4, we present

PF15461 as a case study of functions inferred using predicted structures. In Pfam, the function of PF15461 has been annotated as Beta-carotene 15,15′-dioxygenas, which catalyzes the conversion of beta-carotene to retinal (Fig. 4a) [43, 44]. Retinal is a chromophore that binds integral membrane proteins to form rhodopsin, which is a well-studied bacterial photosynthetic protein in marine environment [45, 46]. The structure-based annotations further showed that this family has functional terms including "oxidoreductase activity" (GO:0016491, MF), single-organism metabolic process (GO:0044710, BP), and "respiratory chain" (GO:0070469, CC), with all of the top ten functional templates detected from enzyme activity (Fig. 4b). These data suggest that the proteins in PF15461 should be mostly involved in the photosynthetic pathway as catalytic enzymes.

Based on the Pfam database, PF15461 is mainly composed of bacteria, which consists of 73% of the sequences. At a family level, the vast majority of the families (37 out of 42 detected families) belong to bacteria, except for 2 and 3 other families belonging to Eukarya and archaea respectively (Fig. 4c). When the sequences from the ocean metagenomic data were included, the number of sequences for this family increased from 369 to 14,353 (a 3889% increase of number of sequences), thus enabling its contact-map and 3D structure to be reliably predicted. In addition, a significant difference was detected for the sequences which were aligned to PF15461 in different water layers (Fig. 4d): 11,544 sequences (average $111 \pm 38.2$) were classified in shallow layer samples (104 runs, < 15 m), and only 2809 sequences (average $20 \pm 8.15$) were classified in deep layer (141 runs, > 15 m), which corresponds to a $p$ value = 1.25e−25 in the Wilcox test. As light could not reach the deep layer which led to the differential distribution of this protein, the data are consistent with the insight that the proteins in PF15461 participate in the bacterial photosynthesis pathway.

As illustrated by the association between the species distribution and functional composition of PF15461 and the marine microbial community, it is precisely because

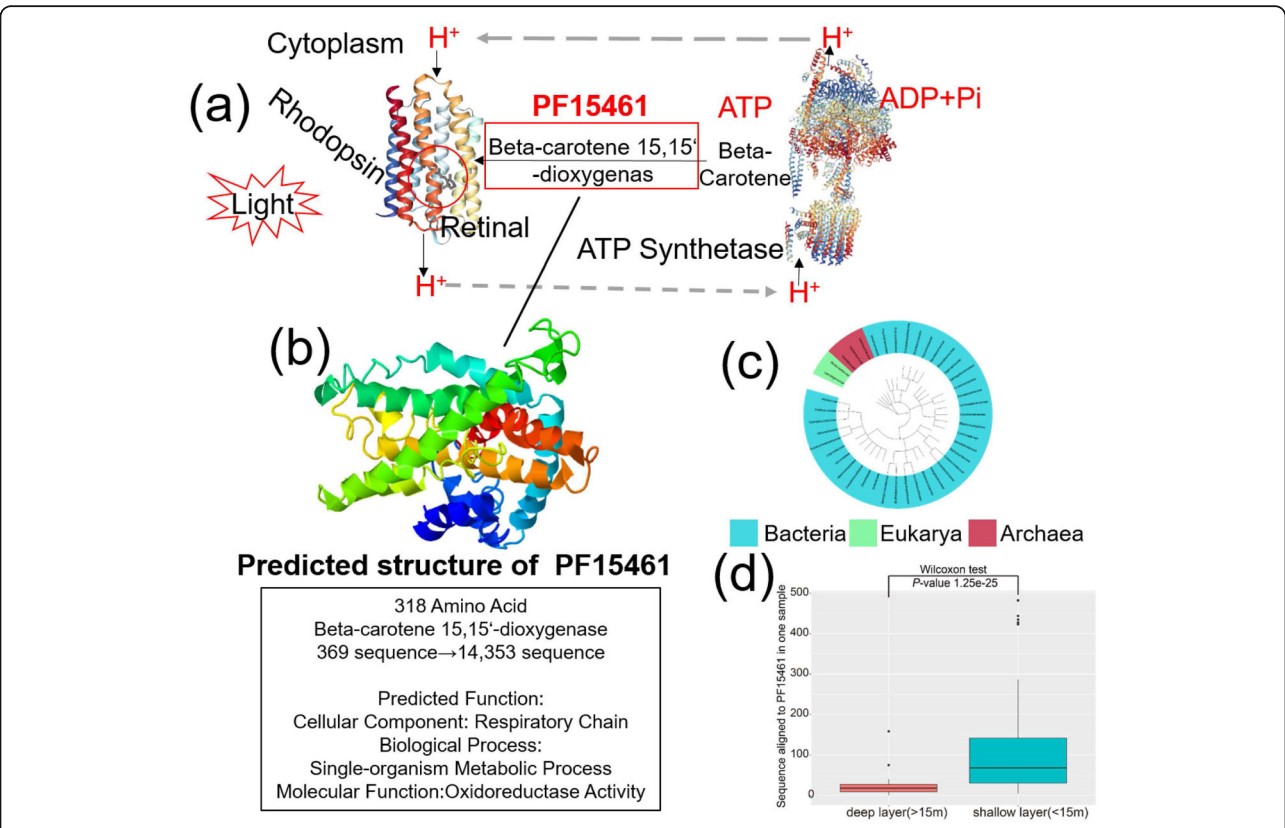

**Fig. 4** Function and species distribution analysis of the PF15461. **a** Proteins in PF15461 participate in the photosynthetic pathways. PF15461 is a Beta-carotene 15,15′-dioxygenas which catalyzes or regulates the conversion of beta-carotene to retinal. **b** Predicted structure and function for PF15461. The structure of PF15461 has a helix-bundle fold, with structure-based function annotations including "oxidoreductase activity" (GO:0016491, MF), single-organism metabolic process (GO:0044710, BP), and "respiratory chain" (GO:0070469, CC). **c** Species distribution on the family level of PF15461. Forty-two families were detected, and the kingdoms to which they belong were marked in different colors. **d** Sequence distribution for PF15461 in different water layers. In *Tara* Oceans dataset, samples were divided in shallow and deep layers and significant differences were detected in two water layers based on Wilcox test

members in PF15461 play critical roles in photosynthesis pathway that the marine microbiome can specifically assist in the modeling of PF15461. Moreover, the same phenomenon can be seen in other families modeled in Fig. 4. Overall, although PF15461 does not represent the model of the highest confidence in the C-QUARK modeling (see a summary of Fig. 2b), the analysis on this protein family represents a representative example illustrating that the inclusion of the marine metagenomic data can help reveal additional insights in both protein structure prediction and function annotations.

### Modeling of additional Pfam families by combining *Tara* with other metagenome databases

In the previous sections, we focus on a specific set of 27 Pfam families that have significant sequence alignment coverage from the *Tara* Oceans dataset. While this exclusive analysis allows a close inspection of how marine microbiome metagenome can assist protein structure and function modeling, the coverage of protein university by the marine samples from *Tara* is limited. To examine the capacity of metagenome-assisted C-QUARK pipeline in ab initio structure prediction, we merged *Tara* into Meta-Clust [47] to form a unified metagenome protein sequence database. A search of the 5721 unknown Pfam families through the database resulted in 1249 families which have a Nf > 64, where 797 of them have Nff = $(Nf_{Tara + MetaClust + UniRef} - Nf_{UniRef})/Nf_{Tara + MetaClust + UniRef} > 0.5$ relative to UniRef. This latter target set (i.e., 417 after excluding the Ovchinnikov et al. dataset) is much larger than the previous set of 27 targets, because MetaClust has a much more diverse source of both host-associated and environmental samples, which are collected through three databases (IMG, NCBI-SRA, and OM-RGC) [47].

In Additional file 2: Table S7, we list the Pfam functional annotations of all the 797 families enhanced by the *Tara*+MetaClust metagenome datasets. Except for 396 families that have no characterized function, most of the remaining Pfam families possess functions which are commonly present as enzymes or structural constitutes of cellular components. Additional file 1: Figure S5 presents the composition distribution of species involved in the 797 Pfam families, where a total of 68,206 records (most of which were recorded at the species level) are obtained and 71.7% of records belong to Bacteria. Further analysis reveals that most of the bacteria are in phylum *Proteobacteria* (widely distributed in a variety of biomes, and has strong adaptability to all kinds of biomes), and phylum *Firmicute* and *Bacteroidetes* (mainly in host-associated environments) [48], while the occurrence frequencies of the photosynthesis-related species are low, despite their dominance distributions in *Tara* Oceans dataset. Overall, while the inclusion of the comprehensive metagenome dataset can significantly increase the MSA depth and structure

modeling coverage, the diverse species and function distributions limit the interpretation of the results from both taxonomical and functional perspective. Meanwhile, the source biome of sequences in MetaClust is less traceable, which hinders further interpretation of the underlying relationship between microbial and Pfam families; these partly highlight the advantage of utilizing specific metagenome sources (such as *Tara* Oceans) on integrated structure and function prediction and annotations.

Additional file 1: Figure S6 presents a summary of the structural modeling results of 417 of the 797 Pfam families, where the 380 families having models reported in [19] were skipped (see Additional file 1: Figure S3 for breakdown of the Pfam families). Among these 417 targets, 235 (56.4%) of them are predicted by C-QUARK to have a correct fold with estimated TM-score > 0.5 according to Eq. (2), while another 147 (35.3%) targets with estimated TM-score between 0.4 and 0.5 (Additional file 1: Figure S6A). There is an obvious correlation between the estimated TM-score and Nf with Pearson correlation coefficient (PCC = 0.44, Additional file 1: Figure S6B), reflecting the impact of MSA construction on contact-map and ab initio structure predictions. As illustrative examples, Additional file 1: Figure S7 presents 24 representative C-QUARK models with different levels of estimated TM-scores. Similar to the models shown in Fig. 2, there is no clear dependence of the estimated quality of the structure models on the type of secondary structures, as high-confidence models are witnessed for all different types *α*-, *β*-, and *αβ*-proteins (Additional file 1: Figure S7A-B). This is quite different from the traditional ab initio structure prediction in which success only limits to the small *α*-proteins [9]; this is mainly due to the success of deep-learning-based long-range contact predictions whose accuracy does not have specific dependence on the secondary structure type of the target sequences [26, 27].

It is worth noting that the 235 proteins only represent a subset of the proteins anticipated to be foldable using the C-QUARK pipeline assisted with the *Tara*+MetaClust databases, as they are only counted from the simulation results of a set of 417 Pfam families with a high effective number of homologous sequences (i.e., Nf > 64) and with more than half of the Nf contribution from the metagenome sequences (i.e., Nff > 0.5) (see Additional file 1: Figure S3). Considering that many high-Nf proteins have been skipped in this modeling experiment, including the 380 families reported in [19] and 452 families whose Nf mainly contributed from UniRef dataset, the number of foldable proteins will be much larger when applying the pipeline to all the 1249 high-Nf families. Moreover, benchmark and blind tests have demonstrated that the deep-learning-based approaches can often create reasonable contact-maps even with a low number of homologous sequences [26, 49]. The application of the pipeline on other low-Nf proteins should also help increase the yield.

Among the 417 modeled targets, 33 proteins are from the same Pfam families as the 227 proteins with models released by Michel et al. in their structure modeling study based on PconsFold2 [20]. Somewhat unexpectedly, there are only 3 targets in which the first models by the C-QUARK and PconsFold2 pipelines have a similar fold with TM-score > 0.5, where the average TM-score between the two is only 0.348 (Additional file 1: Table S8). Even if we count the top five models from each pipeline, the average TM-score of the closest models only marginally increases to 0.403 (data not shown), suggesting diversities of the predicted structures in this small protein dataset. We also listed the estimated TM-score of the C-QUARK models, as well as the model's satisfaction rate of the top-L/5 long-range contacts predicted by the individual pipelines, where there is no obvious correlation of the model similarity between two pipelines with the estimated TM-score or contact satisfaction rates. It is however noticeable that the C-QUARK models have generally a higher contact satisfaction rate (0.476 vs. 0.285 on average with a $p$ value = 1.2e−3 in Student's $t$ test); this difference is probably due to the replica-exchange Monte Carlo simulations implemented by C-QUARK which perform a more extensive (and more time-consuming) conformational search and therefore result in models with a closer match with the predicted contact-maps, compared to the distance geometry simulated annealing method implemented in CNS and PconsFold2 [20].

## Conclusion

Oceanic microbiome contains a large number of novel proteins that are unique for adapting the marine environments. Given recent advancements in sequence-based contact prediction and the contact-guided ab initio folding simulations, the new oceanic microbiome datasets can be used for boosting the accuracy and capacity of protein structure and function predictions. In this study, we utilized more than 1 TB of the oceanic microbiome sequencing data from the *Tara* Oceans project, sifting nearly 100 million predicted protein sequences. The hmmsearch search shows that 2801 out of the 5721 known Pfam families can have at least one member to be homologous with the new *Tara* Oceans sequences, suggesting that the ocean sequences can have potential impact on nearly half of the unknown proteins for computational structure and function predictions. Moreover, a combination of the *Tara* Oceans data with the widely used UniProt sequences can help detect homologous sequences with a Nf score > 64 for 757 Pfam families that have no solved structures, where 444 of them are new relative to that achieved by using the Integrated Microbial Genomes (IMG) dataset [18, 19].

As an illustrative application, we extended C-QUARK, a cutting-edge contact-guided ab initio structure prediction method, to fold the 27 new Pfam families for which

the Tara dataset increased the Nf score by > 50%. Under the guidance of the sequence-based contact predictions, C-QUARK was able to fold 20 (or 74%) of the Pfam targets with an estimated TM-score > 0.5, a percentage consistent with the success rate in the blind CASP13 experiments in which C-QUARK generate correct fold for 33 out of 45 FM targets [38]. Built on the C-QUARK structure models, functional annotations were created by the structure-based function prediction algorithm, MetaGO [42], where a case study on the PF15461 protein family revealed important functions of the proteins in retinal conversion and photosynthesis.

Overall, these studies demonstrated potential usefulness of the new oceanic metagenomic data on protein structure prediction and function annotations. Although the modeling study in this work mainly focused on a subset of 27 (or 417) Pfam families with the largest impact from the *Tara* Oceans (or MetaClust+*Tara*) genome data, the extension on other Pfam families, for which *Tara* and other metagenome datasets help to significantly increase the depth of the multiple sequence alignments, is straightforward. Moreover, recent blind testing experiments have shown that the new contact-map prediction methods, when coupled with convolutional deep-learning networks, are able to generate high-accuracy contact-map for many of the protein targets that have no or very few homologous sequences [27, 38, 49]. Thus, this protocol can be probably applicable to all the unknown Pfam families in order to fully explore the significant impact of the oceanic and other metagenome sequencing. The work along this line is under progress.

## Materials and methods
### Materials

The *Tara* Oceans project is one of the largest collections for marine samples [41, 50]. We obtained 245 whole-genome sequence runs hosted on EBI Metagenomics (ERP001736) (https://www.ebi.ac.uk/metagenomics/studies/ERP001736). These samples were collected from 65 collecting stations, which covered the world's major seas with collection depth ranging from 5 to 1000 m. All samples were derived from data corresponding to size fractions for prokaryotes (0.22–1.6 m or 0.22–3 m) and processed using the EBI Metagenomics portal (now MGnify, https://www.ebi.ac.uk/metagenomics/pipelines/2.0) before downloading.

### *Tara* Oceans raw data process

To obtain predicted genes with high quality and high accuracy, a pipeline for processing the large volume of metagenome was designed. A de novo assembler MEGAHIT v1.0 was used to assemble reads to contigs [51]. Reads in different datasets were assembled individually. For assembling large volume of metagenome, option "--meta-large"

was used, with the minimum length of contigs set to 500 nucleotides. Then, Prodigal (version 2.6) was used to identify open reading frames (ORFs) [52]. For getting high-quality ORFs, options "-c" and "-m" were added to prevent genes from running off edges and avoid building genes across runs of N. In addition, option "-p" was set as meta to predict ORFs from metagenome data rather than single complete genome. CD-HIT v4.6 was used to cluster identical ORFs in each sample [53], and the identity threshold for sequence clustering was set to 95% and the alignment must cover at least 90% of the shorter sequence.

For taxonomy annotation on each sample, 16S rDNA sequence reads were extracted from processed reads using Parallel-Meta v3.2.1 [54] with the "-extract-rna" parameter. The files containing the 16S rDNA sequences (Fasta format) were used as input data and submitted to Parallel-Meta. By aligning non-chimeric reads to the Greengenes database (v13_5) [55], the OTUs were obtained based on a sequence similarity cutoff of 97%. Sensitive alignment mode and Fwd & Rev. pair-end sequence orientation were used. Other parameters were kept default.

### Procedures of the multiple sequence alignments

The procedures of the multiple sequence alignment consist of five consecutive steps as depicted in Fig. 5. At the first step, the default Pfam HMM models of the Pfam families are searched through the *Tara* Oceans sequences by the hmmsearch program from the HMMER package version 3.1b2 [56], where the *Tara* Oceans

sequences with an *E* value ≤ 0.01 and coverage ≥ 75% are selected as "Pfam *Tara* sequences" (Fig. 5a). Second, the Pfam HMM models are searched again by hmmsearch through a combined sequence set of the Pfam *Tara* sequences and UniRef100 [36] to construct MSAs, with option of "-o /dev/null --noali --notextw --cpu 48 --incT 27 -T 27 -A output.sto." For each MSA, Nf score is computed, where the families with Nf ≥ 64 are selected as "effective Pfam families" (Fig. 5b). Third, the Pfam HMM models of the effective Pfam families are searched against *Tara* and the SWISS-PROT [57] by hmmsearch; the sequences with the lowest *E* value from *Tara*/ SWISS-PROT are selected as the "representative sequences of the effective Pfam families" (Fig. 5c). Fourth, a new HMM model is constructed for each representative sequence by the HHblits program from HHsuite version 3.0.0 [58] using option of "-n 8 -e 1E-20 -maxfilt 200000 -neffmax 20 -all"; this program is run through the clustered UniProt database to generate an initial MSA. After filtering these MSAs with HHfilter ("-id 90 -cov 75"), hmmbuild is used to build the new HMM model (Fig. 5d). Finally, the constructed HMM models are searched through a combined database of UniRef100 and Pfam *Tara* sequences by hmmsearch to produce final MSAs. Here, HHfilter (-id 90 -cov 75) is used again to filter the MSAs and the sequences with gap ≥ 75 in the filtered MSA are removed. These final MSAs are used for the next step of contact-map predictions (Fig. 5e). When other metagenome datasets such as

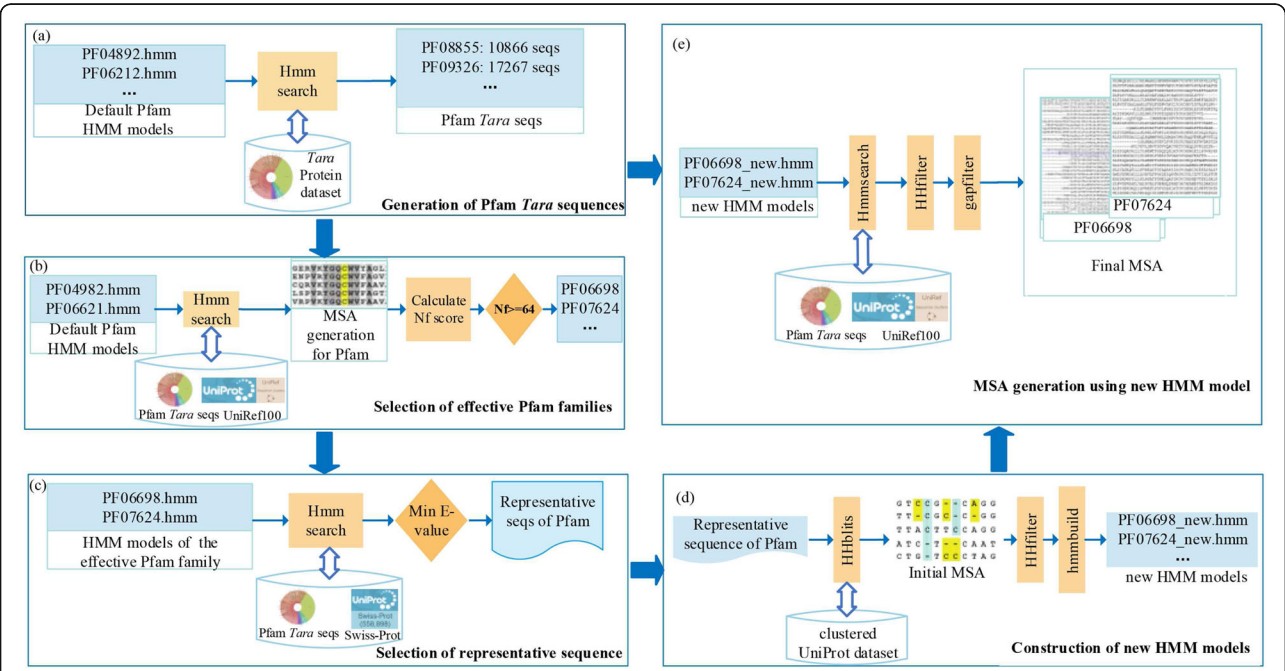

**Fig. 5** Flowchart of multiple sequence alignment construction. (**a**) Generation of Pfam *Tara* sequences. (**b**) Selection of effective Pfam families. (**c**) Selection of representative sequence of Pfam families. (**d**) Construction of new HMM models. (**e**) Final MSA generation using the new HMM model

MetaClust is included, the same procedure is used but with *Tara* Oceans replaced by the extended meta-genome datasets.

### Contact-assisted ab initio structure prediction by C-QUARK

Based on the MSAs, residue-residue contact-maps are predicted using ten state-of-the-art predictors, including NeBcon [13], ResPRE [26], DeepPLM [49], DeepCov [59], Deepcontact [60], DNCON2 [61], MetaPSICOV2 [37], GREMLIN [24], CCMpred [62], and FreeContact [63]. Here, ResPRE and DeepPLM are two in-house contact predictors trained by the deep convolutional neural network based on precision matrix and pseudo-likelihood maximization respectively [26, 49].

Next, the query Pfam sequences are scanned through a set of non-redundant high-resolution PDB structures by gapless threading to generate position-specific fragment structures with continuous length ranging from 1 to 20 AA. A distance profile counting for all distances between a residue pair $i$ and $j$ is derived from the top 200 fragments at each position. This distance profile and the consensus contacts selected from the ten contact predictors are used as restraints, along with the inherent physics-based QUARK force field, to guide C-QUARK to assemble the fragments into full-length structure models through replica-exchange Monte Carlo simulations [64]. Here, to select the contact-maps, the ten predictors are classified into four categories by their prediction accuracy on the training proteins: "very high" (NeBcon, ResPRE, and DeepPLM), "high" (DeepCov, Deepcontact, and DNCON2), "medium" (MetaPSICOV2), and "low" (GREMLIN, CCMpPred, and FreeContact). The consensus contacts are collected from top L, L/2, L/4.5, and L/7.5 contacts from the four categories if Nf < 50 (or from top L, L/2, L/4.5, and L/7.5 contacts from the four categories if Nf ≥ 50). These contacts are implemented into C-QUARK simulation through the following potential:

$$E_{\text{contact}}(d_{ij}) = \begin{cases} -U_{ij}, d_{ij} \le 8 \\ -\dfrac{U_{ij}}{2}\left[1-\sin\left(\dfrac{d_{ij}-(8+d_2)/2}{d_2-8}\pi\right)\right], 8 < d_{ij} \le d_2 \\ \dfrac{U_{ij}}{2}\left[1+\sin\left(\dfrac{d_{ij}-(80+d_2)/2}{80-d_2}\pi\right)\right], d_2 < d_{ij} < 80 \\ U_{ij}, d_{ij} \ge 80 \end{cases}$$

$$(1)$$

where $d_{ij}$ is the $C\beta$ distance between residue pair $i$ and $j$, $U_{ij}$ is the contact prediction confidence score for this residue pair, and $d_2$ is a protein length-dependent

parameter to change the gradient of the well which ranges from 14 to 24 Å.

The decoy conformations from the C-QUARK simulation trajectories are clustered by SPICKER [65] and refined by FG-MD [66]. The model refined from the centroid of the largest cluster is chosen as the first model.

### Benchmark of C-QUARK and quality estimation of predicted models

To estimate the model quality, we benchmark the C-QUARK pipeline on a set of 187 non-homologous proteins that have known structures in PDB, where 111 targets are "hard" and 76 are "easy" according to LOMETS classification [39]. The data in Fig. 6 demonstrate a strong correlation between the TM-score of the C-QUARK models and the confidence score (C-score) of the folding simulations, which has a PCC = 0.813. Here, the confidence score is defined by

$$\text{C-score} = 0.2 \times \ln(\text{Nf}) + \ln(Sr \times Dc) \qquad (2)$$

where Nf is the number of the effective sequences in the MSA. $Sr$ is the weighted satisfaction rate of top-$L$ long-range contacts in the final model, i.e., $Sr = 1/n_L \sum_{i=1}^{n_L} \delta_i w_i^2$, where $n_L$ is the number of the top-$L$ predicted contacts with residue separation > 24, $w_i$ is the weight of the $i$th contact used in C-QUARK which is proportional to the confidence score of the contact-map predictions, and $\delta_i = 1$ (or 0) if the $i$th contact is satisfied (or not satisfied) in the final C-QUARK model. $Dc$ in Eq. (2) measures the degree of structure convergence in the C-QUARK assembly simulation which is calculated by $Dc = \frac{M}{M_{\text{tot}}}/\langle R \rangle$, where $M$ is the number of decoys in the SPICKER cluster, $M_{\text{tot}}$ is the total number of structure decoys generated in C-QUARK simulation, and $\langle R \rangle$ is the average RMSD of the structure decoys to the cluster centroid. The line in Fig. 6 is the fitting equation of data obtained by linear regression:

$$\text{TM-score} = 0.0659 \times \text{C-score} + 0.477 \qquad (3)$$

which has a fitting $\chi^2 = 0.009$. This relation can be used for approximate estimations of the quality of the C-QUARK predicted models, where the root mean square deviation of the TM-score estimation is 0.084.

### Taxonomical and functional analysis for Pfam family

For all microbiome samples, species distribution on genus level was obtained from the corresponding family in the Pfam database. The frequency distribution of these species is then counted, based on which genera with high frequencies were selected, and their

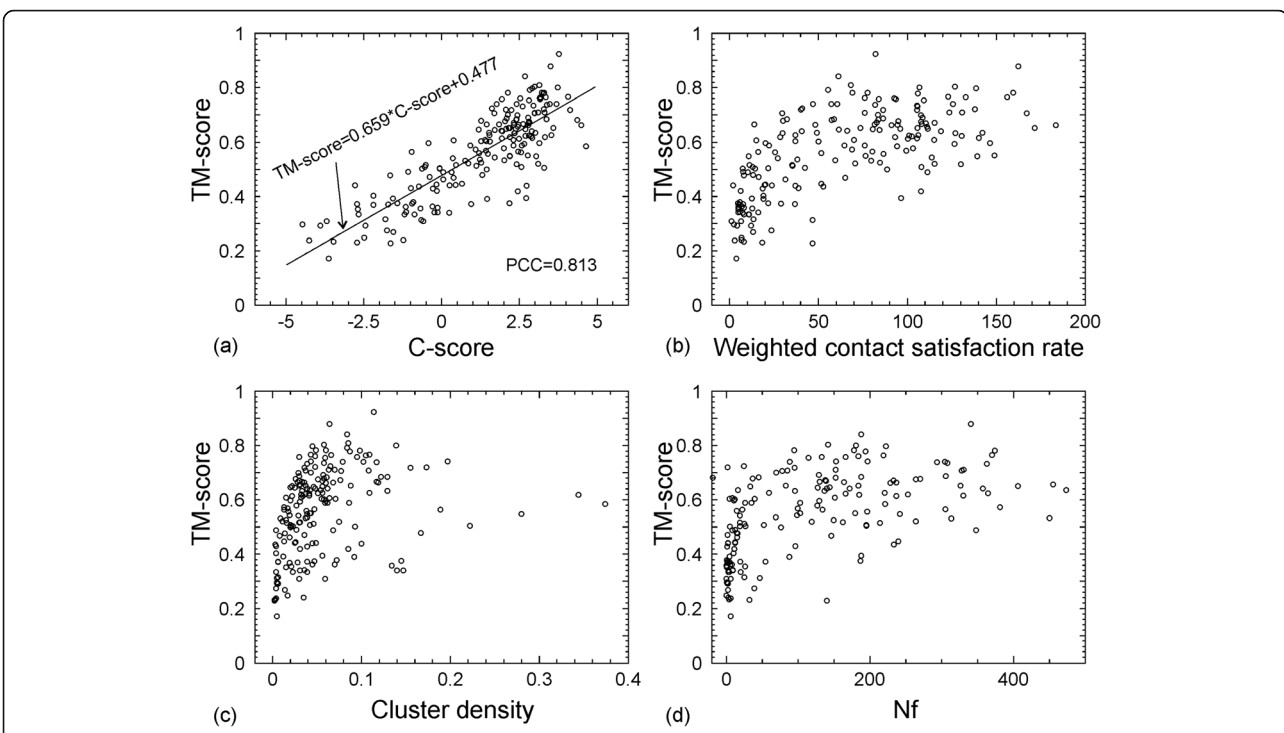

**Fig. 6** Correlation of TM-score of the final models and the parameters of C-QUARK simulations on a set of 187 non-redundant proteins. **a** C-score. **b** Weighted satisfaction rate of contact-map. **c** Structure density of SPICKER cluster. **d** Number of effective sequences in MSA

evolutionary relationship was investigated. PhyloT (http://phylot.biobyte.de/) was used to map the high-frequency species to the NCBI common tree TY 0 (https://www.ncbi.nlm.nih.gov/Taxonomy/Common-Tree/wwwcmt.cgi), and the results were subsequently visualized and modified by an online tool iTOL (assessed as of June/2018).

Given that our data only contains the prokaryotic species of the oceanic organisms, only prokaryotes that occur at high frequencies are taken into calculation. For each dataset, sequence reads for 16S rDNA genes were extracted from processed reads using Parallel-Meta v3.2.1 [54]. By aligning non-chimeric reads to the Greengenes database (v13_5) [55], the OTUs were obtained based on a sequence similarity cutoff of 97%. After all genera identified, their corresponding relative abundance could be normalized and calculated for the 21 bacterial genera.

To deduce the function of predicted structures for the 27 families, MetaGO [42], which is a tool to predict Gene Ontology distribution of proteins by combining sequence homology-based annotation with low-resolution structure prediction and comparison, was performed. Based on the most reliable prediction results (the highest scoring annotations in MetaGO), all of the results were assigned on level-3 Gene Ontology annotations for comparison, according to three GO term categories: Molecular Function, Biological Process, and Cellular Component.

## Supplementary information

**Additional file 1: Figure S1.** Microbial community profiles at phylum and genus levels in 132 datasets. **Figure S2.** Scattering plot of Nf values for 2,801 Pfam families searched through the IMG+Uniref100 versus that searched through the Tara+UniRef100 datasets. **Figure S3.** A breakdown of the Pfam families based on different metagenome database searches. **Figure S4.** Taxonomical distribution of all the genera in the Pfam families that are modellable using different sequence samples. **Figure S5.** Species distribution for 797 Pfam families modeled with the combined Tara and MetaClust dataset. **Figure S6.** Summary of predicted models on 417 Pfam families by C-QUARK using a combined Tara Oceans and MetaClust metagenome dataset. **Figure S7.** Representative C-QUARK structure models predicted using a combined Tara Oceans and MetaClust sequence dataset. **Table S1.** A breakdown of the samples in the IMG database on Feb 21, 2017. **Table S4.** Detailed information of Gene distribution for Tara oceans dataset. **Table S4.** Summary of Nf score and TM-score for the 27 Pfam families modeled. **Table S6.** Structure-based function annotations on 27 Pfam families selected. **Table S8.** Comparison between the first models predicted by C-QUARK and PconsFold2 on a common set of 33 Pfam families.

**Additional file 2: Table S2.** Detailed information of Gene distribution for Tara oceans dataset. **Table S3.** Nf scores calculated based on the UniRef and Tara oceans datasets. **Table S5.** Frequency of 235 genera appeared in the 27 Pfam families. **Table S7.** Pfam functional annotation for 797 families selected by the combined Tara and MetaClust dataset.

**Additional file 3.** Review history.

## Acknowledgements

We thank Dr. Wei Zheng for the technical assistance in IMG/M data preparation. This work used the Extreme Science and Engineering Discovery Environment (XSEDE), which is supported by the National Science Foundation grant number [ACI1548562].

## Review history
The review history is available as Additional file 3.

## Authors' contributions
ZX, KN, and YZ conceived the idea; YW, QS, PY, and CZ performed the study; YW, QS, PY, CZ, and SMM prepared and analyzed the data; YW, KN, and YZ wrote the manuscript. All authors read and approved the final manuscript.

## Funding
This work was supported in part by China's Ministry of Science and Technology's high-tech (863) [2018YFC0910502 to K.N.], National Science Foundation of China [31671374 to K.N.; 61772217 to Y.W. and Z.X.], Fundamental Research Funds for the Central Universities of China [2016YXMS104 to Z. X.], the NIGMS [GM083107, GM116960 to Y.Z.], NIAID [AI134678 to Y.Z.], and the National Science Foundation [DBI1564756 to Y.Z.].

## Availability of data and materials
The *Tara* Ocean dataset is publicly available on EBI Metagenomics (ERP001736) (https://www.ebi.ac.uk/metagenomics/studies/ERP001736). The Metaclust dataset [47] is available as FASTA formatted file at https://metaclust.mmseqs.com/2017_05/. The data from IMG database is available at https://img.jgi.doe.gov/ [67].
The 3D structure files (including multiple sequence alignments, structure and function prediction models), the processed *Tara* Oceans sequence, and the modeling data for all unknown Pfam families assisted with the *Tara* Oceans and MetaClust datasets are available at both https://zhanglabs.github.io/Tara-3D/ [68] and https://zhanglab.ccmb.med.umich.edu/Tara-3D/. Pipeline script source codes are available at GitHub [68] and Zenodo [69] under the MIT license.

## Ethics approval and consent to participate
Not applicable.

## Consent for publication
All authors have approved the manuscript for submission.

## Competing interests
The authors declare that they have no competing interests.

## Author details
<sup>1</sup>College of Life Science and Technology and College of Software, Huazhong University of Science and Technology, Wuhan 430074, Hubei, China. <sup>2</sup>Department of Computational Medicine and Bioinformatics, University of Michigan, Ann Arbor, MI 48109, USA. <sup>3</sup>Department of Biological Chemistry, University of Michigan, Ann Arbor, MI 48109, USA.

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

## 

