## [**Additional file 3.** Review history. · Genome Biology]

Review History

First round of review

Reviewer 1

Are you able to assess all statistics in the manuscript, including the appropriateness of statistical tests used? Yes, and I have assessed the statistics in my report.

Comments to author:

Summary:

The authors explore an excited new source of metagenomic sequences that are not currently part of the IMG/JGI metagenomic database and find that it enriches certain protein families that were previously not accessible to 3D modeling. The authors go on to apply their latest 3D modeling protocol to make models and annotate 20/27 protein families that have a significant boost in the number of sequences upon addition of tara-ocean data.

General comment:

Though I do not doubt that the tara ocean project samples areas of the ocean that are not currently available in IMG/JGI aquatic sequence metagenomic database... the statement that IMG database is mostly from human/animal guts may be misleading.

Detailed comments

Abstract

"where 20 have a correct fold with TM-score >0.5"

Do you mean "where 20 are predicted to have the correct fold"? Unless some of these have since been experimentally confirmed?

Page 4 lines 34-35

"IMG database which has the sequence samples mainly from human and animal guts, where the ocean microbiome, one of the richest sources of organisms on this planet... has been neglected"

A large fraction of sequences in the IMG/JGI database are aquatic. Saying that the Ocean sequences are "neglected" does not seem accurate. IMG database today suggests their "aquatic" database is the largest they have. See the following link:

<https://img.jgi.doe.gov/cgi-bin/m/main.cgi>

Table S1 reports a significantly higher number for human host-associated samples compared to the website.... not clear if there was a large removal of sequences recently or an error in the table. Either way... if authors want to claim that the previous study by Ovchinnikov et al. did not use aquatic/ocean sequences, statistics from their date of collection should be reported. Perhaps a scatter plot comparing Nf values from the previous study to this study can be shown.

In Figure S2, the authors compare the IMG to tara-ocean databases and state: "Based on taxonomical database and literature review, the phylum Cyanophyta (labeled green in two panels) which is dominant in the ocean microbiome are more prevalent in the Tara oceans dataset than in IMG database."

This seems a bit misleading, as it would make more sense to compare actual counts and not the fractions. The IMG hosts a wide-range of metagenomic projects from many different groups, if the authors want to make the statement that tara-ocean provides a different source (or more of a specific source), they should only compare to the ocean/aquatic databases hosted at IMG.

Page 6, lines 33-34

"Contrast to the IMG database which is mainly from human and animal guts, 65% of the genera in Tara belong to Prokaryotes,"
This part is a little confusing... Isn't most of human/animal gut microbiome also made up of prokaryotes?

Some missing citations:

Page 4, lines 21-22: "which are mainly attributed to the accuracy improvement of sequence-based contact-map predictions [10-13]" should also include reference describing the initial pseudo-likelihood and the mfDCA approach :

<https://www.ncbi.nlm.nih.gov/pubmed/21268112>

<https://www.pnas.org/content/108/49/E1293>

The authors don't seem to mention MetaClust, which was a recent effort to assemble and cluster all metagenomic databases, including those of IMG/JGI and tara-ocean. For the readers, it would be useful to know the difference in methodology and if the method to assemble reads in the current manuscript, provides any additional sequences.

<https://metaclust.mmseqs.org/>

Reviewer 2

Are you able to assess all statistics in the manuscript, including the appropriateness of statistical tests used? There are no statistics in the manuscript.

Comments to author:

In this paper the authors use metagenomics from marine samples to enhance the number of sequences for a number of Pfam families. Then the authors use their pipeline to predict the structure of 27 Pfam families. In principle the study is well made, but the results are not surprising. There are one study (by Baker) that run a similar pipeline a few years ago for all Pfam families using Gremlin. We have also run modeling for all Pfam families and presented them last year. Both these studies presented about 500 high quality models each (and not only 27). On the other hand these 27 models are most likely of a higher quality than presented earlier.

My main problem with the paper is however that the method does not seem to be available for me to run locally, nor is the dataset of sequences. Today in the era of open science this is not acceptable to this reviewer.

A few other major questions are:

1. Why only marine samples ? Why not use all metagenomic samples?
2. Why not run the pipeline for all pipelines
3. Is really the 64 efficient sequences an efficient cutoff. In our experience there is a la
4. Why do the authors only release the 27 models and not the results for all Pfam families. Also failed predictions are of interest to the community.
5. The datasets and methods should be available (the sequence database could be of interest to others)
6. A comparison with earlier models for all Pfam families would be valuable.

Minor:

1. It is very confusing that the authors report "TM-score" although it is just an estimated TM-score. This should be changed in many places.

Response to Reviewer #1

We very much appreciate the comments and suggestions from the Reviewer. One of the major concerns from the Review is on the misleading citation of the aspartic samples in IMG dataset. This was due to the statistics on an earlier IMG version collected when we started the project in 2017. We have added statistics from the newest IMG data in Table S1 and had the outdated citation removed. The Reviewer also concerned on our ignorance of other important and comprehensive metagenome databases, where we have performed new calculations on the MetaClust dataset with results discussed in a newly added section. We also fixed a number of citation and typo issues pointed out by the Reviewer. Overall, we found that the Reviewer's comments are very help to improve our work and manuscript. Below, we include point-by-point replies to the comments of the Reviewer, where all changes have been highlighted in yellow in the manuscript.

1. The Reviewer comments that

Summary:

The authors explore an excited new source of metagenomic sequences that are not currently part of the IMG/JGI metagenomic database and find that it enriches certain protein families that were previously not accessible to 3D modeling. The authors go on to apply their latest 3D modeling protocol to make models and annotate 20/27 protein families that have a significant boost in the number of sequences upon addition of tara-ocean data.

Response:

We thank the Reviewer for positive comments on the work.

2. The Reviewer comments that

General comment:

Though I do not doubt that the tara ocean project samples areas of the ocean that are not currently available in IMG/JGI aquatic sequence metagenomic database... the statement that IMG database is mostly from human/animal guts may be misleading.

Page 4 lines 34-35

*"IMG database which has the sequence samples mainly from human and animal guts, where the ocean microbiome, one of the richest sources of organisms on this planet... has been neglected" A large fraction of sequences in the IMG/JGI database are aquatic. Saying that the Ocean sequences are "neglected" does not seem accurate. IMG database today suggests their "aquatic" database is the largest they have. See the following link:
<https://img.jgi.doe.gov/cgi-bin/m/main.cgi>*

Response:

We thank the Reviewer for pointing out this important point. Our former statement was made on the IMG data in 2017 when this project was initiated; this was also the dataset close to what the Ovchinnikov *et al.* used. At that time, 52.57% of all 17,054 samples in IMG were from human and animal gut microbiome, where only 18.86% samples (sum to 1.7 billion genes without clustering) are from aquatic biomes (Table S1).

But we agree with the Reviewer that that the new data in IMG contain more aquatic samples.

To clarify the issue, we added the statistics of the latest IMG dataset in Table S1, in which the percentages for gut microbiome of Human and Animal and aquatic biomes became 37.18 % and 35.96 % respectively. Accordingly, we have removed the statement to avoid potential confusion (see Page 2).

3. The Reviewer comments that

Detailed comments

Abstract "where 20 have a correct fold with TM-score >0.5"

Do you mean "where 20 are predicted to have the correct fold"? Unless some of these have since been experimentally confirmed?

Response:

Yes, we meant TM-score estimated based on our confidence score system which have been stringently tested based on large-scale benchmark experiments. To clarify the point, we have changed "TM-score" to "estimated TM-score" in Abstract, as well as all relevant sentences throughout the manuscript.

4. The Reviewer comments that

Table S1 reports a significantly higher number for human host-associated samples compared to the website.... not clear if there was a large removal of sequences recently or an error in the table. Either way... if authors want to claim that the previous study by Ovchinnikov et al. did not use aquatic/ocean sequences, statistics from their date of collection should be reported. Perhaps a scatter plot comparing Nf values from the previous study to this study can be shown.

Response:

As mentioned in Point-1, the former Table S1 was based on the statistics of the IMG data on Feb 21, 2017 when the project was started. This was also the dataset close to what Ovchinnikov *et al.* used. In the new version, we added the statistics of the latest IMG collected on June 5, 2019, which indeed showed a significant increase in the number of samples for all the species. Accordingly, the percentage of aquatic biomes was increased from 18.86% to 35.86%.

Following the suggestion, we also added a new figure (Figure S2) to compare the Nf values from IMG and *Tara* Oceans datasets. Although IMG has generally a higher Nf value due to the comprehensiveness of metagenome datasets, there are a high number of families (201 out of 2,801 cases) in which Nf >64 only occur in the *Tara* Ocean dataset, showing that the *Tara* is complementary to the IMG dataset. We added the following paragraph in Page 3 to discuss the data:

In **Fig S2**, we show a head-to-head comparison of Nf values of all the 2,801 Pfam families calculated on the *Tara* Ocean and IMG datasets that are combined separately with UniRef100. Although the use of IMG generally results in higher Nf values, there are a considerable number of cases (201) in which Nf >64 occurs only when the *Tara* Oceans dataset is used, indicating the complementarity of the *Tara* Oceans to the IMG dataset for assisting the modeling of different Pfam families.

5. The Reviewer comments that

In Figure S2, the authors compare the IMG to tara-ocean databases and state: "Based on taxonomical database and literature review, the phylum Cyanophyta (labeled green in two panels) which is dominant in the ocean microbiome are more prevalent in the Tara oceans dataset than in IMG database."

This seems a bit misleading, as it would make more sense to compare actual counts and not the fractions. The IMG hosts a wide-range of metagenomic projects from many different groups, if the authors want to make the statement that tara-ocean provides a different source (or more of a specific source), they should only compare to the ocean/aquatic databases hosted at IMG.

Response:

Thank you for raising the issue. We believe there is a misunderstanding here, due to our misleading figure caption of the original Figure S2 (now Figure S4). In fact, the purpose of this figure is not to directly compare the IMG and *Tara* samples and demonstrate that "tara-ocean provides a different source". Instead, the figure tries to examine the taxonomical distribution of the genera in the Pfam families that are modellable using different genome samples. Here, 'modellable' refers to the cases with $N_f > 64$ using the IMG set by Ovchinnikov *et al.* or the case with $N_f > 64$ and $N_{ff} > 0.5$ using *Tara Oceans* set in this study. The assumption is that the genus distribution in the Pfam families should be correlated with the genus distribution in the metagenome samples which were used to model the Pfam families, due to the homologous relation between sequences detected in the MSA construction tools.

This assumption seems confirmed by the data presented in the figure, i.e., phyla (*Bacteroidetes* and *Firmicutes*) which are common in the gut microbiome account for the majority in the Pfam families modellable using the IMG, while phyla (such as *Cyanophyta*) which are common in the ocean microbiome are more prevalent in the Pfam families modellable using the *Tara Oceans* dataset. In this context, we suggest that the introduction of new genome samples can help increase the coverage of specific modellable Pfam families.

To clarify the confusion, we have rewritten the caption of Fig S4 (former Fig S2) as following:

Figure S4. Taxonomical distribution of all the genera in the Pfam families that are modellable using different sequence samples. Here, 'modellable' refers to the cases with $N_f > 64$ using the IMG set by Ovchinnikov *et al.* or the case with $N_f > 64$ and $N_{ff} > 0.5$ using *Tara Oceans* set in this study. (A) using IMG database with 614 Pfam families; (B) using *Tara Oceans* with 27 Pfam families. Different colors represent different phylum, and the bar corresponding to the outer circle represents the count of the Pfam families where the species was detected. Phylum *Bacteroidetes* and *Firmicutes* (labeled red in two panels), which are common in the gut microbiome according to taxonomical database and literature review, account for the overwhelming majority in the Pfam families modellable using the IMG. The phylum *Cyanophyta* (labeled green in two panels), which is dominant in the ocean microbiome, are more prevalent in the Pfam families modellable using the *Tara Oceans* dataset.

6. The Reviewer comments that

Page 6, lines 33-34

"Contrast to the IMG database which is mainly from human and animal guts, 65% of the genera in Tara belong to Prokaryotes,"

This part is a little confusing... Isn't most of human/animal gut microbiome also made up of prokaryotes?

Response:

We apologize for the typo. The word ‘Prokaryotes’ was supposed to be ‘Proteobacteria’, which is the dominate phylum in environmental microbiome. We have corrected the typo accordingly (see Page 4).

7. The Reviewer comments that

Some missing citations:

Page 4, lines 21-22: "which are mainly attributed to the accuracy improvement of sequence-based contact-map predictions [10-13]" should also include reference describing the initial pseudo-likelihood and the mfDCA approach:

<https://www.ncbi.nlm.nih.gov/pubmed/21268112>

<https://www.pnas.org/content/108/49/E1293>

Response:

We have included these two references in the citation. Thank you.

8. The Reviewer comments that

The authors don't seem to mention MetaClust, which was a recent effort to assemble and cluster all metagenomic databases, including those of IMG/JGI and tara-ocean. For the readers, it would be useful to know the difference in methodology and if the method to assemble reads in the current manuscript, provides any additional sequences.

<https://metaclust.mmseqs.org/>

Response:

This is a good point. To address this issue, we added a new section to discuss the modeling results based on MetaClust when combined with *Tara* and UniRef databases (see Page 5-6):

Modeling of additional Pfam families by combining *Tara* with other metagenome databases

In the previous sections, we focus on a specific set of 27 Pfam families that have significant sequence alignment coverage from the *Tara* Oceans dataset. While this exclusive analysis allows a close inspection of how marine microbiome metagenome can assist protein structure and function modeling, the coverage of protein universality by the marine samples from *Tara* is limited. To examine the capacity of metagenome-assisted C-QUARK pipeline in *ab initio* structure prediction, we merged *Tara* into MetaClust [47] to form a unified metagenome protein sequence database. A search of the 5,721 unknown Pfam families through the database resulted in 1,249 families which have a $N_f > 64$, where 797 of them have $N_{ff} = (N_{f_{Tara+MetaClust+UniRef}} - N_{f_{UniRef}}) / N_{f_{Tara+MetaClust+UniRef}} > 0.5$ relative to UniRef. This latter target set (i.e., 417 after excluding the Ovchinnikov *et al* dataset) is much larger than the previous set of 27 targets, because MetaClust has a much more diverse source of both host-associated and environmental samples, which are collected through three databases (IMG, NCBI-SRA, and OM-RGC) [47].

In **Table S8**, we list the Pfam functional annotations of all the 797 families enhanced by the *Tara*+MetaClust metagenome datasets. Except for 396 families that have no characterized function, most of the remaining Pfam families possess functions which are commonly present as enzymes or structural constituents of cellular components. **Fig S5** presents the composition distribution of species involved in the 797 Pfam families, where a total of 68,206 records (most of which were recorded at the species level) are obtained and 71.7% of records belong to Bacteria. Further analysis reveals that most of the bacteria are in phylum *Proteobacteria* (widely distributed in a variety of biomes, and has strong adaptability to all kinds of biomes), and phylum *Firmicute* and *Bacteroidetes* (mainly in host-associated environments) [48], while the occurrence frequencies of the photosynthesis-related species are low, despite their dominance distributions in *Tara* Oceans

dataset. Overall, while the inclusion of the comprehensive metagenome dataset can significantly increase the MSA depth and structure modeling coverage, the diverse species and function distributions limit the interpretation of the results from both taxonomical and functional perspective. Meanwhile, the source biome of sequences in MetaClust is less traceable, which hinders further interpretation of the underlying relationship between microbial and Pfam families; these partly highlight the advantage of utilizing specific metagenome sources (such as *Tara Oceans*) on integrated structure and function prediction and annotations.

Fig S6 presents a summary of the structural modeling results of 417 of the 797 Pfam families, where the 380 families having models reported in [19] were skipped (see **Fig S3** for breakdown of the Pfam families). Among these 417 targets, 235 (56.4%) of them are predicted by C-QUARK to have a correct fold with estimated TM-score >0.5 according to Eq. (2), while another 147 (35.3%) targets with estimated TM-score between 0.4 and 0.5 (**Fig S6A**). There is an obvious correlation between the estimated TM-score and Nf with Pearson correlation coefficient (PCC=0.44, **Fig S6B**), reflecting the impact of MSA construction on contact-map and *ab initio* structure predictions. As illustrative examples, **Fig S7** presents 24 representative C-QUARK models with different levels of estimated TM-scores. Similar to the models shown in **Fig 2**, there is no clear dependence of the estimated quality of the structure models on the type of secondary structures, as high confidence models are witnessed for all different types α -, β -, and $\alpha\beta$ -proteins (**Fig. S7A-B**). This is quite different from the traditional *ab initio* structure prediction in which success only limits to the small α -proteins [9]; this is mainly due to the success of deep-learning based long-range contact predictions whose accuracy does not have specific dependence on the secondary structure type of the target sequences [26, 27].

It is worth noting that the 235 proteins only represent a subset of the proteins anticipated to be foldable using the C-QUARK pipeline assisted with the *Tara*+MetaClust databases, as they are only counted from the simulation results of a set of 417 Pfam families with a high effective number of homologous sequences (i.e. Nf >64) and with more than half of the Nf contribution from the metagenome sequences (i.e., Nff >0.5) (see **Fig S3**). Considering that many high-Nf proteins have been skipped in this modeling experiment, including the 380 families reported in [19] and 452 families whose Nf mainly contributed from UniRef dataset, the number of foldable proteins will be much larger when applying the pipeline to all the 1,249 high-Nf families. Moreover, benchmark and blind tests have demonstrated that the deep-learning based approaches can often create reasonable contact-maps even with a low number of homologous sequences [26, 49]. The application of the pipeline on other low-Nf proteins should also help increase the yield.

Among the 417 modelled targets, 33 proteins are from the same Pfam families as the 227 proteins with models released by Michel *et al.* in their structure modeling study based on PconsFold2 [20]. Somewhat unexpectedly, there are only 3 targets in which the first models by the C-QUARK and PconsFold2 pipelines have a similar fold with TM-score >0.5 , where the average TM-score between the two is only 0.348 (Table S7). Even if we count the top-five models from each pipeline, the average TM-score of the closest models only marginally increases to 0.403 (data not shown), suggesting diversities of the predicted structures in this small protein dataset. We also listed the estimated TM-score of the C-QUARK models, as well as the model's satisfaction rate of the top-L/5 long-range contacts predicted by the individual pipelines, where there is no obviously correlation of the model similarity between two pipelines with the estimated TM-score or contact satisfaction rates. It is however noticeable that the C-QUARK models have generally a higher contact satisfaction rate (0.476 vs. 0.285 on average with a p -value=1.2E-3 in Student's t-test); this difference is probably due to the replica-exchange Monte Carlo simulations implemented by C-QUARK which perform a more extensive (and more time-consuming) conformational search and therefore result in models with a closer match with the predicted contact-maps, compared to the distance geometry simulated annealing method implemented in CNS and PconsFold2 [20].

Response to Reviewer #2

We very much appreciate the comments from the Reviewer, which we found very helpful for improving the quality of our work. One concern from the Reviewer is on the absence of our sequence data and method. In the revision, we have now released all the data and scripts, as well as the C-QUARK server. We plan to release a standalone C-QUARK package after the C-QUARK method paper is published and with the programs finalized. The Reviewer also concerned about the completeness the modeling targets and the comprehensiveness of the metagenome datasets, where we have now extended the pipeline to model a substantially larger set of Pfam families based on the combined *Tara* and MetaClust metagenome samples. We also made several other clarifications to address the issues raised by the Reviewer. In the following, we include point-by-point replies to the comments of the Reviewer, where all changes have been highlighted in yellow in the manuscript

1. The Reviewer comments that

In this paper the authors use metagenomics from marine samples to enhance the number of sequences for a number of Pfam families. Then the authors use their pipeline to predict the structure of 27 Pfam families. In principle the study is well made, but the results are not surprising. There are one study (by Baker) that run a similar pipeline a few years ago for all Pfam families using Gremlin. We have also run modeling for all Pfam families and presented them last year. Both these studies presented about 500 high quality models each (and not only 27). On the other hand these 27 models are most likely of a higher quality than presented earlier.

- 1. Why only marine samples? Why not use all metagenomic samples?*
- 2. Why not run the pipeline for all pipelines*
- 4. Why do the authors only release the 27 models and not the results for all Pfam families. Also failed predictions are of interest to the community.*

Response:

Thank you for the important comments. First, the 27 Pfam families discussed in the paper do not represent the total number of proteins that are modellable using our pipeline. In fact, one of the major purposes of this study was to examine the specific impact of the marine samples on the gain of the modeling of new Pfam families, and meanwhile to interpret the relationship between the Pfam families predicted by the microbiome data and their functional role in the marine environment. We have thus focused only on the 27 Pfam families that have $N_f > 64$ and with more than half of the homologous sequences from the *Tara* Ocean data (i.e., $N_{ff} > 0.5$); meanwhile skipping the targets that were already reported by Baker and co-workers based on the IMG samples. We believe that our description in the former manuscript was not clear, which have resulted in the misunderstanding. To help clarify the point, we added a new figure (Fig S3) in SI which outlines the pipeline of the Pfam family selection.

Second, we agree that it is important to have a complete test of the current pipeline for the structure modeling of new Pfam families, based on more comprehensive metagenome datasets. For this purpose, we performed a new experiment by extending C-QUARK to model 417 unknown Pfam families based on a combined *Tara* and MetaClust dataset, which have resulted in 235 new Pfam families that are expected to have correct fold (with an estimated TM-score > 0.5). All models of the 417 proteins, together with former 27 Pfam

families, are released in the Tara-3D: <https://zhanglab.ccmb.med.umich.edu/Tara-3D/>.

Nevertheless, as pointed out below, the 417 Pfam families represent still only a subset of proteins modellable by the current pipeline, while the application of the pipeline to all 5,721 proteins is straightforward and ongoing, but takes a longer time. The modeling results on the whole set of the unknown Pfam families will be added to the Tara-3D webpage after the simulations are complete.

In the corrected manuscript, we added a new section to discuss the results and limit of the new experiments (see Pages 5-6):

Modeling of additional Pfam families by combining *Tara* with other metagenome databases

In the previous sections, we focus on a specific set of 27 Pfam families that have significant sequence alignment coverage from the *Tara* Oceans dataset. While this exclusive analysis allows a close inspection of how marine microbiome metagenome can assist protein structure and function modeling, the coverage of protein universality by the marine samples from *Tara* is limited. To examine the capacity of metagenome-assisted C-QUARK pipeline in *ab initio* structure prediction, we merged *Tara* into MetaClust [47] to form a unified metagenome protein sequence database. A search of the 5,721 unknown Pfam families through the database resulted in 1,249 families which have a $Nf > 64$, where 797 of them have $Nff = (Nf_{Tara+MetaClust+UniRef} - Nf_{UniRef}) / Nf_{Tara+MetaClust+UniRef} > 0.5$ relative to UniRef. This latter target set (i.e., 417 after excluding the Ovchinnikov *et al* dataset) is much larger than the previous set of 27 targets, because MetaClust has a much more diverse source of both host-associated and environmental samples, which are collected through three databases (IMG, NCBI-SRA, and OM-RGC) [47].

In **Table S8**, we list the Pfam functional annotations of all the 797 families enhanced by the *Tara*+MetaClust metagenome datasets. Except for 396 families that have no characterized function, most of the remaining Pfam families possess functions which are commonly present as enzymes or structural constituents of cellular components. **Fig S5** presents the composition distribution of species involved in the 797 Pfam families, where a total of 68,206 records (most of which were recorded at the species level) are obtained and 71.7% of records belong to Bacteria. Further analysis reveals that most of the bacteria are in phylum *Proteobacteria* (widely distributed in a variety of biomes, and has strong adaptability to all kinds of biomes), and phylum *Firmicute* and *Bacteroidetes* (mainly in host-associated environments) [48], while the occurrence frequencies of the photosynthesis-related species are low, despite their dominance distributions in *Tara* Oceans dataset. Overall, while the inclusion of the comprehensive metagenome dataset can significantly increase the MSA depth and structure modeling coverage, the diverse species and function distributions limit the interpretation of the results from both taxonomical and functional perspective. Meanwhile, the source biome of sequences in MetaClust is less traceable, which hinders further interpretation of the underlying relationship between microbial and Pfam families; these partly highlight the advantage of utilizing specific metagenome sources (such as *Tara* Oceans) on integrated structure and function prediction and annotations.

Fig S6 presents a summary of the structural modeling results of 417 of the 797 Pfam families, where the 380 families having models reported in [19] were skipped (see **Fig S3** for breakdown of the Pfam families). Among these 417 targets, 235 (56.4%) of them are predicted by C-QUARK to have a correct fold with estimated TM-score > 0.5 according to Eq. (2), while another 147 (35.3%) targets with estimated TM-score between 0.4 and 0.5 (**Fig S6A**). There is an obvious correlation between the estimated TM-score and Nf with Pearson correlation coefficient (PCC=0.44, **Fig S6B**), reflecting the impact of MSA construction on contact-map and *ab initio* structure predictions. As illustrative examples, **Fig S7** presents 24 representative C-QUARK models with different levels of estimated TM-scores. Similar to the models shown in **Fig 2**, there is no clear dependence of the estimated quality of the structure models on the type of secondary structures, as high confidence models are witnessed for all different types α -, β -, and $\alpha\beta$ -proteins (**Fig. S7A-B**). This is quite different from the traditional *ab initio* structure prediction in which success only limits to the small α -proteins [9]; this is mainly due to the success of deep-learning based long-range contact predictions whose accuracy does not have specific dependence on the secondary structure type of the target sequences [26, 27].

It is worth noting that the 235 proteins only represent a subset of the proteins anticipated to be foldable using the C-QUARK pipeline assisted with the *Tara*+MetaClust databases, as they are

only counted from the simulation results of a set of 417 Pfam families with a high effective number of homologous sequences (i.e. $N_f > 64$) and with more than half of the N_f contribution from the metagenome sequences (i.e., $N_{ff} > 0.5$) (see **Fig S3**). Considering that many high- N_f proteins have been skipped in this modeling experiment, including the 380 families reported in [19] and 452 families whose N_f mainly contributed from UniRef dataset, the number of foldable proteins will be much larger when applying the pipeline to all the 1,249 high- N_f families. Moreover, benchmark and blind tests have demonstrated that the deep-learning based approaches can often create reasonable contact-maps even with a low number of homologous sequences [26, 49]. The application of the pipeline on other low- N_f proteins should also help increase the yield.

Among the 417 modelled targets, 33 proteins are from the same Pfam families as the 227 proteins with models released by Michel *et al.* in their structure modeling study based on PconsFold2 [20]. Somewhat unexpectedly, there are only 3 targets in which the first models by the C-QUARK and PconsFold2 pipelines have a similar fold with TM-score > 0.5 , where the average TM-score between the two is only 0.348 (Table S7). Even if we count the top-five models from each pipeline, the average TM-score of the closest models only marginally increases to 0.403 (data not shown), suggesting diversities of the predicted structures in this small protein dataset. We also listed the estimated TM-score of the C-QUARK models, as well as the model's satisfaction rate of the top-L/5 long-range contacts predicted by the individual pipelines, where there is no obviously correlation of the model similarity between two pipelines with the estimated TM-score or contact satisfaction rates. It is however noticeable that the C-QUARK models have generally a higher contact satisfaction rate (0.476 vs. 0.285 on average with a p -value=1.2E-3 in Student's t-test); this difference is probably due to the replica-exchange Monte Carlo simulations implemented by C-QUARK which perform a more extensive (and more time-consuming) conformational search and therefore result in models with a closer match with the predicted contact-maps, compared to the distance geometry simulated annealing method implemented in CNS and PconsFold2 [20].

2. The Reviewer comments that

My main problem with the paper is however that the method does not seem to be available for me to run locally, nor is the dataset of sequences. Today in the era of open science this is not acceptable to this reviewer.

5. The datasets and methods should be available (the sequence database could be of interest to others)

Response:

Following the suggestion, the Tara sequence dataset and structural models, together with the source codes of scripts for creating MSA and data analyses, are made publicly available now at <https://zhanglab.ccmb.med.umich.edu/Tara-3D/>. The on-line server of the C-QUARK pipeline is also available at <https://zhanglab.ccmb.med.umich.edu/C-QUARK/>. The complete method of C-QUARK is described in another method paper, which is currently in revision and some components need to be finalized. We plan to release a standalone version of C-QUARK after the method paper is published with the programs finalized.

3. The Reviewer comments that

3. Is really the 64 efficient sequences an efficient cutoff. In our experience there is a la

Response:

This is a good point and the selection of $N_f=64$ is empirical. According to our structure modeling results on the FM targets in CASP13, although C-QUARK could generate acceptable models for a few targets with $N_f < 64$, the cutoff of $N_f=64$ can split the FM domains into two groups with significantly different TM-score on average (0.49 vs. 0.67,

corresponding approximately the minimum p -value=0.001). Meanwhile, the former study by Ovchinnikov *et al.* also selected the same cutoff, where the consistent Nf cutoff helped facilitate the comparison of both studies. We summarized the insight in the following paragraph (see Page 3):

Here we note that the selection of the Nf cutoff (=64) is empirical. In fact, as per analysis of 45 FM domains in CASP13, although our pipeline can generate correct fold with TM-score >0.5 for several targets with Nf <64, the overall quality and success rate for the targets with Nf >64 are much higher [38]. Approximately, a cutoff of Nf=64 splits the FM targets into two groups with the average TM-score (0.49 vs. 0.67) that corresponds to the lowest p -value (=0.001) in Student's t-test. We therefore continue to use this cutoff, consistent with the previous study of Ovchinnikov *et al.* [19].

4. The Reviewer comments that

6. *A comparison with earlier models for all Pfam families would be valuable.*

Response:

Since our purpose of the study was to examine the potential of our pipeline, when combined with the *Tara* Oceans database, for modeling new Pfam families, we have selected to skip the families with models reported by Ovchinnikov *et al.*. Thus, we could not compare our models directly with those of Ovchinnikov *et al.*, due to the lack of overlapped protein sets.

On the other hand, the Elofsson lab has recently published an independent study of Pfam family structure prediction by the combination of PconsC3 contact prediction with the CONFOLD folding simulations (with the latter built on the CNS program). There are a set of 33 Pfam families that are commonly shared by the Elofsson and our studies. We have added the following paragraph to summarize the comparison of the structural models from both pipelines (see Page 6):

Among the 417 modelled targets, 33 proteins are from the same Pfam families as the 227 proteins with models released by Michel *et al.* in their structure modeling study based on PconsFold2 [20]. Somewhat unexpectedly, there are only 3 targets in which the first models by the C-QUARK and PconsFold2 pipelines have a similar fold with TM-score >0.5, where the average TM-score between the two is only 0.348 (Table S7). Even if we count the top-five models from each pipeline, the average TM-score of the closest models only marginally increases to 0.403 (data not shown), suggesting diversities of the predicted structures in this small protein dataset. We also listed the estimated TM-score of the C-QUARK models, as well as the model's satisfaction rate of the top-L/5 long-range contacts predicted by the individual pipelines, where there is no obviously correlation of the model similarity between two pipelines with the estimated TM-score or contact satisfaction rates. It is however noticeable that the C-QUARK models have generally a higher contact satisfaction rate (0.476 vs. 0.285 on average with a p -value=1.2E-3 in Student's t-test); this difference is probably due to the replica-exchange Monte Carlo simulations implemented by C-QUARK which perform a more extensive (and more time-consuming) conformational search and therefore result in models with a closer match with the predicted contact-maps, compared to the distance geometry simulated annealing method implemented in CNS and PconsFold2 [20].

5. The Reviewer comments that

Minor:

1. *It is very confusing that the authors report "TM-score" although it is just an estimated TM-score. This should be changed in many places.*

Response:

We have replaced 'TM-score' by 'estimated TM-score' throughout the manuscript. Thank you!